

# New Constraints on Biogenic Emissions using Satellite-Based Estimates of Carbon Monoxide Fluxes

Helen M. Worden[1], A. Anthony Bloom[2], John R. Worden[2], Zhe Jiang[3], Eloise Marais[4], Trissevgeni Stavrakou[5], Benjamin Gaubert[1], Forrest Lacey[1]

[1]Atmospheric Chemistry Observations and Modelling (ACOM), National Center for Atmospheric Research (NCAR), Boulder, CO, USA
[2]Jet Propulsion Laboratory, California Institute of Technology, Pasadena, CA, USA
[3]School of Earth and Space Sciences, University of Science and Technology of China, Hefei, China
[4]Department of Physics and Astronomy, University of Leicester, Leicester, UK
[5]Royal Belgian Institute for Space Aeronomy (BIRA-IASB), Brussels, Belgium

*Correspondence to*: Helen Worden (hmw@ucar.edu)

**Abstract.** Biogenic non-methane volatile organic compounds (NMVOCs) emitted from vegetation are a primary source for the chemical production of carbon monoxide (CO) in the atmosphere and these biogenic emissions account for about 18% of the global CO burden. Partitioning CO fluxes to different source types in top-down inversion methods is challenging and typically a simple scaling of the posterior flux to prior flux values for fossil fuel, biogenic and biomass burning sources is used. Here we show top-down estimates of biogenic CO fluxes using a Bayesian inference approach, which explicitly accounts for both posterior and a priori CO flux uncertainties. This approach re-partitions CO fluxes following inversion of Measurements Of Pollution In The Troposphere (MOPITT) CO observations with the GEOS-Chem model, a global chemical transport model driven by assimilated meteorology from the NASA Goddard Earth Observing System (GEOS). We compare these results to the prior information for CO used to represent biogenic NMVOCs from GEOS-Chem, which uses the Model of Emissions of Gases and Aerosols from Nature (MEGAN) for biogenic emissions. We evaluate the a posteriori biogenic CO fluxes against top-down estimates of isoprene fluxes using Ozone Monitoring Instrument (OMI) formaldehyde observations. We find similar seasonality and spatial consistency in the posterior CO and top-down isoprene estimates globally. For the African savanna region, both top-down CO and isoprene seasonality vary significantly from the MEGAN apriori inventory. This method for estimating biogenic sources of CO will provide an independent constraint on modelled biogenic emissions and has the potential for diagnosing decadal-scale changes in emissions due to land-use change and climate variability.

## 1 Introduction

Carbon monoxide (CO) plays a critical role in tropospheric chemistry and climate as a precursor to greenhouse gases ozone ($O_3$) and carbon dioxide ($CO_2$) and through its influence on methane ($CH_4$) lifetime via its destruction by the hydroxyl radical (OH) (e.g., IPCC AR5: Myhre et al, 2013; Gaubert et al., 2017). CO is formed in the atmosphere from direct emission during incomplete combustion of biomass and fossil fuels and from the oxidation of hydrocarbons. Biogenic non-methane volatile organic compounds (NMVOCs) emitted from vegetation represent a significant source of precursors that oxidize and



produce CO, accounting for around 18% of the global CO budget (e.g., Folberth et al., 2006, Table 8, not including anthropogenic VOCs). Duncan et al. (2007) calculated a contribution of photochemically produced CO from biogenic NMVOC sources contributes about 15% of total CO sources. Pfister et al. (2008) showed that oxidation from isoprene (C5H8) alone contributes to 9 to 16 % of the global CO burden, with a global yield of CO from isoprene of 0.30, calculated on a per carbon basis, where CO production is more efficient in polluted environments (i.e. high NOx). Most biogenic NMVOC emissions have relatively short atmospheric lifetimes, typically < 1 hour, so that transport away from sources is negligible (e.g., Palmer et al., 2003). This allows the estimation of primary NMVOC emissions (e.g., isoprene) using secondary products such as formaldehyde (HCHO), that can be more easily observed with remote sensing (e.g., Palmer et al., 2003, Stavrakou et al., 2009a,b, Marais et al., 2012, Bauwens et al., 2016). Biogenic CO is then produced from HCHO and other NMVOCs through photolysis and reactions with OH, where HCHO lifetime is on the order of hours in tropical daytime (e.g., Miller et al., 2008, Anderson et al., 2017). The chemical production and transport of CO away from sources must be modeled using chemical transport models (CTMs) within an inversion framework. Previous efforts to estimate the amount of atmospheric CO that is produced chemically from biogenic NMVOC emissions have used MOPITT (Measurements of Pollution in The Troposphere) satellite observations as a "top-down" constraint while estimating CO fluxes from different sectors such as fossil fuels, biomass burning and biogenic NMVOCs (Fortems-Cheiney et al., 2011; Hooghiemstra et al., 2011, 2012; Yin et al., 2015; Jiang et al., 2017). These estimates have updated the prior fluxes in these sectors. However, if the prior fluxes relied on inventories with inaccurate assumptions about relative partitioning and seasonal variability, these errors are propagated into the posterior emission estimates.

The Model of Emissions of Gases and Aerosols from Nature (MEGAN, Guenther et al., 2006) and other models of biogenic emissions such as Organizing Carbon and Hydrology in Dynamic EcosystEms (ORCHIDEE, Krinner et al., 2005) have made significant strides in allowing a more accurate representation of these emissions in chemical transport models (CTMs). However, evaluation and testing of these models is challenging due to limited availability of correlative measurements, especially in tropical regions where biogenic emissions are largest. Comparisons of CTMs using MEGAN have been performed with surface and airborne in situ observations of isoprene and other biogenic NMVOCs with reasonable agreement such as in the Southeast U.S., (e.g., Warneke et al., 2010), but these are only over limited regional scales. Large scale evaluation of biogenic emission models has relied on satellite observations of HCHO to constrain top-down isoprene emission estimates globally (e.g., Shim et al., 2005; Stavrakou et al., 2009b, Bauwens et al., 2016); and regionally for North America (e.g., Palmer et al., 2003, 2006; Millet et al., 2008), Southeastern Asia (Fu et al., 2007), South America (Barkley et al., 2008), Europe (Dufour et al., 2009; Curci et al., 2010) and Africa (Marais et al., 2012; 2014).

The ability to accurately model and predict biogenic emissions has become increasingly important as trade-offs in land use are studied for potential climate change mitigation (e.g., Griscom et al., 2017; Luyssaert et al., 2018). These trade-offs include carbon uptake, albedo changes and the emissions of biogenic VOCs. Since biogenic emissions are precursors to both positive



(ozone and methane) and negative (secondary organic aerosols) climate forcers, there is significant uncertainty in their role (e.g., Unger et al., 2014a,b; Scott et al., 2018; Harper et al., 2018;  Luyssaert et al., 2018). The results presented here for CO from biogenic NMVOC sources give additional, independent information from global satellite observations that can be used to constrain biogenic emissions in areas that are not well monitored with other measurements.

**2 CO flux estimation**

The basis for estimates of CO flux from biogenic sources is a 15-year inversion analysis (Jiang et al, 2017) that used the adjoint of the GEOS-Chem model (Henze et al., 2007) and MOPITT Version 6J multispectral CO observations (Deeter et al., 2014). This approach used latitude bias-corrected MOPITT data (total CO columns and CO vertical profiles) averaged on the GEOS-Chem 5° longitude x 4° latitude grid to constrain model estimates of monthly CO fluxes in each grid cell from three primary source sectors: anthropogenic fossil fuel and biofuel, biomass burning and oxidation from BVOCs. CO from methane oxidation, ~28% of the global CO budget (Folberth et al., 2006), was estimated to be 877 Tg(CO/yr as an aggregated global source. The Model of Emissions of Gases and Aerosols from Nature (MEGAN), version 2.0 (Guenther et al., 2006) was used to formulate the prior CO emissions from BVOCs. Biomass burning prior fluxes are from the Global Fire Emission Database (GFED3; van der Werf et al., 2010) and global prior fluxes for fossil fuel are from the Emission Database for Global Atmospheric Research (EDGAR 3.2FT2000; Olivier and Berdowski, 2001) with updated inventories for the northern hemisphere described in Jiang et al., (2017).

Model errors in atmospheric transport and chemistry typically propagate into the largest sources of uncertainty when quantifying CO fluxes with satellite observations (Jones et al., 2003; Stavrakou et al. 2006; Kopacz et al., 2010; Jiang et al., 2013, Müller et al. 2018). The impact of these errors is reduced in Jiang et al, (2017) by applying an initial assimilation of MOPITT CO over ocean regions to establish boundary conditions that are consistent with the satellite observations before the adjoint emission estimation over land source regions. This approach accounts for CO chemistry and transport over the ocean and allows continental source regions to be treated more independently (Jiang et al. 2015). To characterize remaining errors due to transport in the CO emission estimates, three different inversions are obtained using MOPITT CO total column, full profile and lower troposphere profile retrievals and their corresponding averaging kernels (Jiang et al., 2013, Worden, J. et al. 2013). Since CO total column observations have no vertical information, they are less sensitive to convection and local emission sources, but they provide information on advection and chemistry with better measurement precision than profile data. Vertical profiles of CO, especially when restricted to the lower troposphere contain more information about local sources. However, since these vertical distributions have worse precision, the flux estimates are still impacted by model errors in convection, advection and chemistry. An ensemble covariance of these three inversion results provides an empirical evaluation of the sensitivity in CO fluxes to altitude dependent constraints and their corresponding corrections in the presence of model transport and chemistry errors (Worden et al. 2017). We find the largest variation in the three emission estimates for CO in



India and Indonesia where large sources, strong convection and advection from other regions all contribute significantly. As in the Worden et al. (2017) analysis, we constrain the monthly total CO flux in each 5° longitude x 4° latitude grid box using the mean and variance from the three inversion estimates described above.

One particular limitation of using the inversion results of Jiang et al. (2017) for biogenic CO fluxes is the use of different meteorological data fields over the 2001-2015 period. Due to availability at the time the inversion analysis was conducted, different versions of the NASA Goddard Earth Observing System (GEOS) assimilated meteorological fields were applied: GEOS-4 (2000-2003), GEOS-5 (2004-2012) and GEOS-FP (2013-2015). Since MEGAN uses the meteorological fields as inputs, the different GEOS versions produce non-negligible discontinuities in the a priori for biogenic CO for these time

periods. For this reason, and to overlap with the availability of OMI (Ozone Monitoring Instrument) formaldehyde data for inferring isoprene fluxes, we consider the period from 2005 to 2012 for the analysis presented here.

### 3 Bayesian CO flux attribution approach

We provide a brief overview of the methodology and we refer the reader to the Worden et al. (2017) study for a complete
description of the Bayesian partitioning approach. Monthly estimates of biogenic, (BIO), biomass burning, (BB) and fossil fuel, (FF) CO fluxes—and their associated uncertainties—were calculated via Bayesian inference, where

$$p(BB,BIO,FF|\mathbf{A}) \propto \frac{p(BB,BIO,FF)p(F|\mathbf{A})}{p(F)} \qquad \text{Eq. 1.}$$

$p(BB,BIO,FF)$ and $p(BB,BIO,FF|\mathbf{A})$ are the joint prior and posterior distributions of BIO, BB and FF, A represents the atmospheric CO measurements, p(F) and p(F|A) are the prior and posterior probability distributions of total CO flux $F$ within each monthly 5° x 4° grid box. p(F|A) was empirically approximated based on three CO flux inversion estimates, and p(F) was assumed to be ± 50%. In Worden et al., (2017), the re-partitioned distribution $p(BB,BIO,FF|\mathbf{A})$ was sampled using an adaptive Metropolis-Hastings Markov Chain (MCMC) Monte Carlo algorithm (Bloom et al., 2015). Prior uncertainties for BB were
estimated using emission factor uncertainties for fire types reported for GFED4 and prior uncertainties of ± 50% assumed for BIO and FF. While Jiang et al. (2017) also estimated sector contributions by scaling the a priori fluxes, these estimates account for the full characterization of sectoral uncertainties given both prior and posterior uncertainty estimates. We note that chemical production of CO from methane oxidation (877 Tg(CO/yr from Jiang et al., 2017) is considered a fixed term in the Bayesian attribution due to the longer chemical lifetime of methane and consequent global influence.



## 4 Uncertainty prediction and limitations

Uncertainties are available by 5° x 4° grid cell, month and source sector (BB, FF or BIO) and represent the 1-sigma width of the posterior distributions; these distributions are critically dependent on the a priori uncertainties and therefore subject to change when different a priori distributions and covariances are assumed in the Bayesian attribution approach. One of the

assumptions in this study is the prior uncertainty in BB, which only considers emission factor uncertainties (Akagi et al., 2011) and does not explicitly account for other factors in BB CO fluxes such as combustion completeness and biomass (fuel) amount (e.g. Bloom et al. 2015). On average, for the remote tropical regions we consider in this study, we find monthly grid cell scale posterior errors around 24% for BIO CO, 22% for BB CO and 45% for FF CO, indicating reasonable improvements over prior errors for BIO and BB and only modest FF improvement. While these factors could have large uncertainties in individual grid

cells, errors will partially cancel out when considering larger regions with global trace gas budget constraints. Future work will examine the effects of using a wider range of prior uncertainties that reflect multiple inventories and the estimated errors for all parameters in the bottom-up emissions.

We also note that there is an implicit assumption in the re-partitioning for CO fluxes from biogenic emissions that monthly

time scales and relatively large grid box sizes will account for the chemical production of CO from the primary biogenic emissions within the grid box. This assumption relies on the short (< 1 day) chemical lifetime of most biogenic emissions, especially isoprene and formaldehyde, the accuracy of CO chemistry in GEOS-Chem, and the relatively smaller uncertainties for BB and FF fluxes. However, the large grid boxes could also be a source of error in GEOS-Chem chemistry for the inversion results. Kaiser et al. (2018) showed that finer grid scales (0.25° × 0.3125°) and accurate representations of NOx emissions in

GEOS-Chem produced top-down isoprene estimates from HCHO observations that compared better to aircraft in situ observations. Furthermore, the GEOS-Chem inversions did not consider chemical non-linearities due to changes in OH caused by changing CO emissions (Gaubert et al., 2016). This has led to an overall increase in OH over the decade 2003-2013 and thus is responsible for an overall increase in secondary CO chemical production (Gaubert et al. 2017). Model intercomparisons and scale sensitivity tests would help quantify the uncertainties from these assumptions.

## 5 Top-down isoprene estimates

Since isoprene represents the dominant biogenic NMVOC emission (e.g., Guenther et al., 2006; 2012) and accounts for 66% of biogenic NMVOC emissions that react to produce CO (Folberth et al., 2006) we compare our estimated CO fluxes from biogenic sources with global estimates of isoprene as a way to check their spatial and temporal variability. Here we use the biogenic isoprene emission estimates provided by the GlobEmission project at

http://emissions.aeronomie.be/index.php/omi-based/biogenic. Using OMI satellite observations of tropospheric formaldehyde as a constraint (De Smedt et al., 2015), the GlobEmission estimates of biogenic isoprene emission are





produced on a global 0.5° x 0.5° grid using the adjoint of the IMAGESv2 global chemistry-transport model (Stavrakou et al., 2015, Bauwens et al., 2016) with a priori isoprene emissions from MEGAN-MOHYCAN described in Stavrakou et al. (2014).

Model results for biogenic emissions depend on both static and dynamic input from the CTM and the corresponding meteorology data or reanalysis driving the CTM. Isoprene emissions using MEGAN (Guenther et al., 2006; 2012) are computed as:

$$E_{ISOP} = E_o \times \gamma_{PAR} \times \gamma_T \times \gamma_{AGE} \times \gamma_{SM} \times \gamma_{CE} \qquad\qquad \text{Eq. 2}$$

where $E_o$ is the emission flux under standard conditions, and the $\gamma$ parameters are dimensionless scaling factors that account for sensitivities to temperature ($T$), leaf age distribution ($AGE$), soil moisture ($SM$) and the canopy radiative environment ($CE$). The last term includes the effects of leaf area index (LAI) and the plant sensitivity to the above canopy radiance. Values of $E_o$ are specified in MEGAN using a global database of plant functional types (PFT) assuming 5 PFTs (broadleaf trees, needleleaf

trees, grasses, crops and shrubs). The other parameters require dynamic input such as hourly temperature, wind speed, humidity, solar radiation, soil moisture from the meteorological fields used in the CTM and monthly LAI from the Moderate Resolution Imaging Spectroradiometer (MODIS) on the EOS/Terra and EOS/Aqua satellites. Figure 1 shows 2005-2012 average biogenic CO and isoprene fluxes for 40°S to 40°N as estimated with MEGAN and meteorological data (GEOS-5 for CO and ECMWF for isoprene) as compared to estimated fluxes using top-down constraints from satellite observations

(MOPITT for CO and OMI HCHO for isoprene).

We find that the distributions for biogenic CO follow similar spatial patterns as the isoprene fluxes (albeit coarser spatial resolution) and that the top down estimates are in general lower than the emissions predicted using MEGAN, as found in previous studies (e.g., Millet et al., 2008; Stavrakou et al., 2009a; Marais et al., 2014).

**6 Global budgets of CO and C5H8 from biogenic emissions**

Table 1 shows the annual average fluxes of CO and C5H8 for the 2005-2012 period for 80°S to 80°N, Northern mid-latitudes (20°N to 40°N), Tropics (20°S to 20°N), and the separated tropical regions of South America, Africa, and Maritime Continent. Our global estimate for BIO CO from non-methane sources (566 ± 49 Tg(CO)/yr) is in agreement with a previous estimate (546 Tg(CO)/yr, Folberth et al. (2006)) which was obtained by adding the contributions to CO from isoprene (359), methanol

(110), terpenes (49) and acetone (28). However, we find that BIO CO fluxes are a larger percentage (~40%) of the sum of BB, FF and BIO CO sources than expected based on previous budgets (~27%) (e.g., Folberth et al., 2006, which has 811 Tg(CO)/yr for BB and 672 Tg (CO)/yr for FF). This is largely due to the decreased contribution of BB CO associated with a



decline in tropical fires over this period (e.g., Andela et al., 2017), as well as declining FF CO emissions (Yin et al., 2015; Strode et al., 2016; Jiang et al., 2017; Zheng et al., 2018).

## 7 Seasonality of biogenic emissions – case study for the North African Savannas

Figure 2 shows the seasonal behavior of posterior sectoral CO flux estimates in the N. Africa savannas (see outlined grid cells
of Figure 2 inset map) derived by the Bayesian attribution approach described in section 3. While biomass burning (BB) dominates in N.H. winter, and fossil fuel fluxes (FF) have little variability, biogenic fluxes show two broad maxima, one in April and the other in October. We note that these maxima are not likely mis-identified BB fluxes as the BB months are relatively well defined in the region for November to February.

Figure 3 shows the time series of apriori (MEGAN with GEOS-5) vs. posterior for the N. Africa Savannas region, with surface temperature from the Modern-Era Retrospective analysis for Research and Applications (MERRA, Rienecker et al., 2011) overplotted to show the correspondence of the posterior results with temperature variability. We note that using GEOS-FP for the 2013-2015 meteorology (not shown) results in a ~27% increase for the peak apriori (MEGAN with GEOS-FP) biogenic CO fluxes compared to the years using MEGAN with GEOS-5.

As shown by Marais et al. (2014), the seasonality of isoprene fluxes in the African savannas north of the equator also have a maximum in April followed by a minimum during the rainy season, June to September (Janicot, et al., 2008). Fluxes for December to March were not estimated in Marais et al. (2014) due to interference with biomass burning emissions and secondary formation HCHO. The top-down isoprene estimates from Bauwens et al. (2016) and the CO flux estimates from the
Bayesian attribution approach described here both show 2 minima in biogenic emissions for this region, one in the rainy season (June-August) and the other in winter (December-January) similar to the surface temperature (Fig. 4). Marais et al., (2014) attribute the higher emissions from MEGAN to the model dependence on LAI, which has a broad maximum in August.

Other regions in South America, Southern Africa and Australia that were tested for seasonality of BIO CO fluxes (see
supplementary material) did not show the same large inconsistency with MEGAN, suggesting that the N. African savannas require special treatment and a revised parametrization within MEGAN to account for the enhanced sensitivity to surface temperature vs. LAI during the rainy season. The tabulated emissions under standard conditions, $E_o$, could also require revision to account for human-driven changes in plant types due to cropland expansion in the N. African savannas region in recent decades (e.g., Andela et al., 2014).



### 8 Summary and future work

This paper has presented the first results for estimates of CO from biogenic NMVOCs using a Bayesian re-partitioning of top-down flux estimates. We find that the CO flux estimates based on MOPITT CO observations are spatially consistent with biogenic isoprene flux estimates based on OMI HCHO observations. Both top-down estimates for carbon monoxide and

isoprene suggest that biogenic emissions based on MEGAN are too high in the Tropics by 28% for isoprene and 10% for carbon monoxide with the largest discrepancies in South America. As a case study in tropical North Africa, we found that the top-down estimates suggest a significant seasonality change compared to MEGAN for both CO and C5H8. The top-down estimates have seasonal cycles that match well with MERRA surface temperature and that have secondary minima during the rainy season that are not predicted well by MEGAN. These discrepancies suggest the potential for regional updates to the

MEGAN model, a focus of future work. Sensitivity to model grid scales that affect transport and chemistry uncertainties will also be investigated.

In order to examine climate variability and possible trends in biogenic emissions, the methods described here will also be applied to a flux inversion estimate using a consistent meteorological reanalysis. Since MOPITT will soon have a 20-year data

record, it will span several ENSO (El Niño Southern Oscillation) cycles and will have the potential for detecting the effects of inter-annual and long-term changes in surface temperatures on biogenic CO flux variability.

**Data availability.** MOPITT data sets used for the CO inverse modeling component of this study are publicly available at http://reverb.echo.nasa.gov and at https://eosweb.larc.nasa.gov/datapool. The isoprene emission estimates obtained from

inverse modeling of OMI HCHO observations are available from the GlobEmission project at http://emissions.aeronomie.be/index.php/omi-based/biogenic.

**Author contributions.** HW performed the analysis of flux estimates. AAB performed the Bayesian flux attribution. JW and HW contributed to the design and use of MOPITT data for the CO inverse modeling and flux attribution. ZJ performed the

CO inverse modeling. EM assisted with interpretation of results. TS provided guidance on the use of the isoprene estimates. FL and BJ assisted with interpretation of results. HW, AAB and JW prepared the original manuscript, and all authors contributed to the review and editing of the manuscript.

**Competing interests**. The authors declare that they have no conflict of interest.


**Acknowledgements.** The authors thank Rebecca Schwantes and Ivan Ortega Martinez at NCAR for their helpful review and comments. EAM acknowledges funding from NERC/EPSRC (grant number EP/R513465/1). The MOPITT team acknowledges the contributions of COMDEV and ABB BOMEM with support from the Canadian Space Agency (CSA), the



Natural Sciences and Engineering Research Council (NSERC) and Environment Canada. The NCAR MOPITT project is supported by the National Aeronautics and Space Administration (NASA) Earth Observing System (EOS) Program. The National Center for Atmospheric Research (NCAR) is sponsored by the National Science Foundation. Part of this research was carried out at the Jet Propulsion Laboratory, California Institute of Technology, under a contract with the National Aeronautics and Space Administration.

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



**Tables and Figures**

Table 1. Annual average CO and C5H8 fluxes for 2005-2012

| Region | ECMWF/ MEGAN Isoprene Tg(C5H8)/yr | OMI/BIRA Isoprene Tg(C5H8)/yr | GEOS-Chem/ MEGAN BIO CO Tg(CO)/yr | GEOS-Chem/ MOPITT BIO CO Tg(CO)/yr | GEOS-Chem/ MOPITT BB CO Tg(CO)/yr | GEOS-Chem/ MOPITT FF CO Tg(CO)/yr |
|---|---|---|---|---|---|---|
| Tropics (20S-20N) | 246 | 176 | 364 | $326 \pm 27$ | $231 \pm 14$ | $120 \pm 14$ |
| Tr. S. Amer. (90-30W)· | 127 | 83 | 131 | $104 \pm 20$ | $41 \pm 8$ | $16 \pm 7$ |
| Tr. Africa (20W-50E)· | 73 | 56 | 166 | $159 \pm 36$ | $145 \pm 23$ | $34 \pm 14$ |
| Maritime C. (90-160E)· | 39 | 32 | 57 | $52 \pm 16$ | $43 \pm 9$ | $44 \pm 12$ |
| N. Midlat (20N-40N) | 34 | 28 | 99 | $95 \pm 11$ | $15 \pm 1$ | $264 \pm 18$ |
| Global (80S-80N) | 343 | 273 | 630 | $566 \pm 49$ | $290 \pm 18$ | $534 \pm 42$ |

·Latitude range is 20S-20N




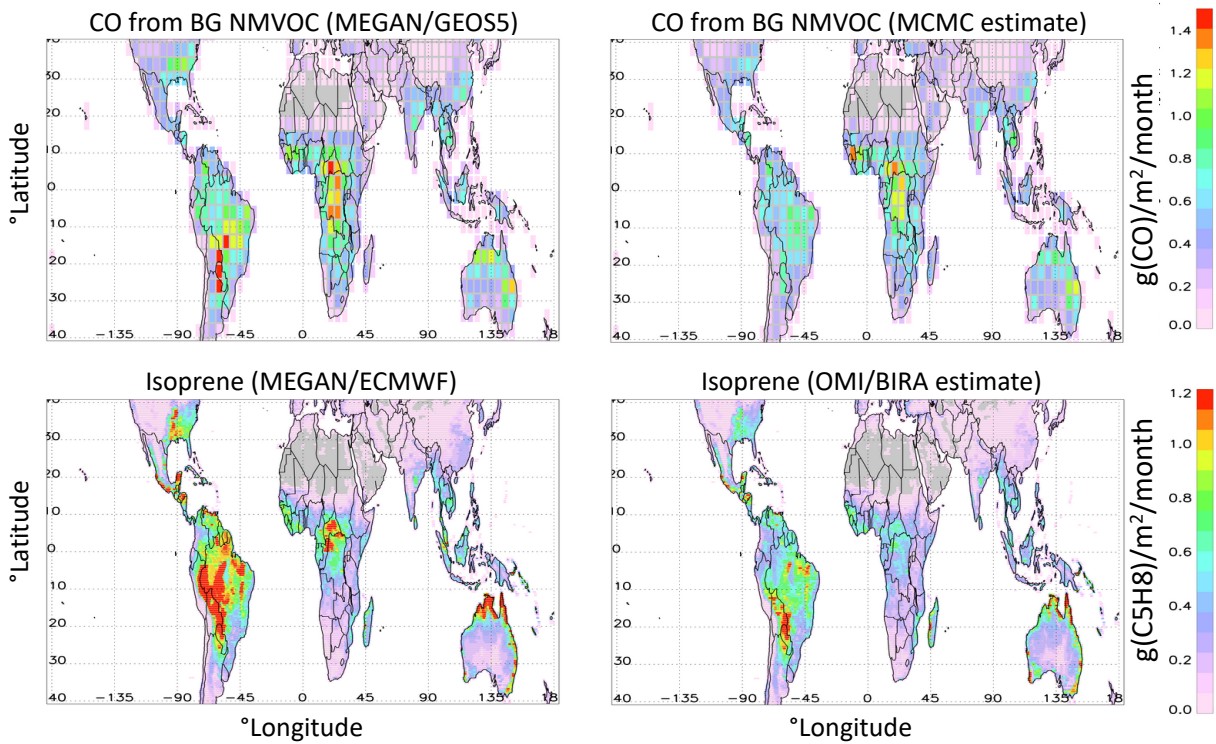

*Figure 1. 2005-2012 average biogenic flux for CO (top) and isoprene (bottom) with model estimates using MEGAN on the left and top-down estimates using MOPITT observations for CO (top right) and isoprene inferred from OMI HCHO observations (bottom left).*





**Figure 2. A posteriori (solid lines) and a priori (dotted lines) CO fluxes averaged for each month over 2005-2012 for the N. Africa savannas region for biomass burning (BB, red), biogenic (BIO, green) and fossil fuel (FF, blue) sectors. The inset map shows average BIO CO fluxes over Africa, with the same color scale as shown in the top panels of Fig. 1. The N. Africa savannas grid boxes considered for the monthly averages are outlined in gray. Errors on the 8-yr average fluxes for this region are indicated for each sector and month, with values around 2.3% for BIO, 1.6% for BB and 3.4% for FF.**





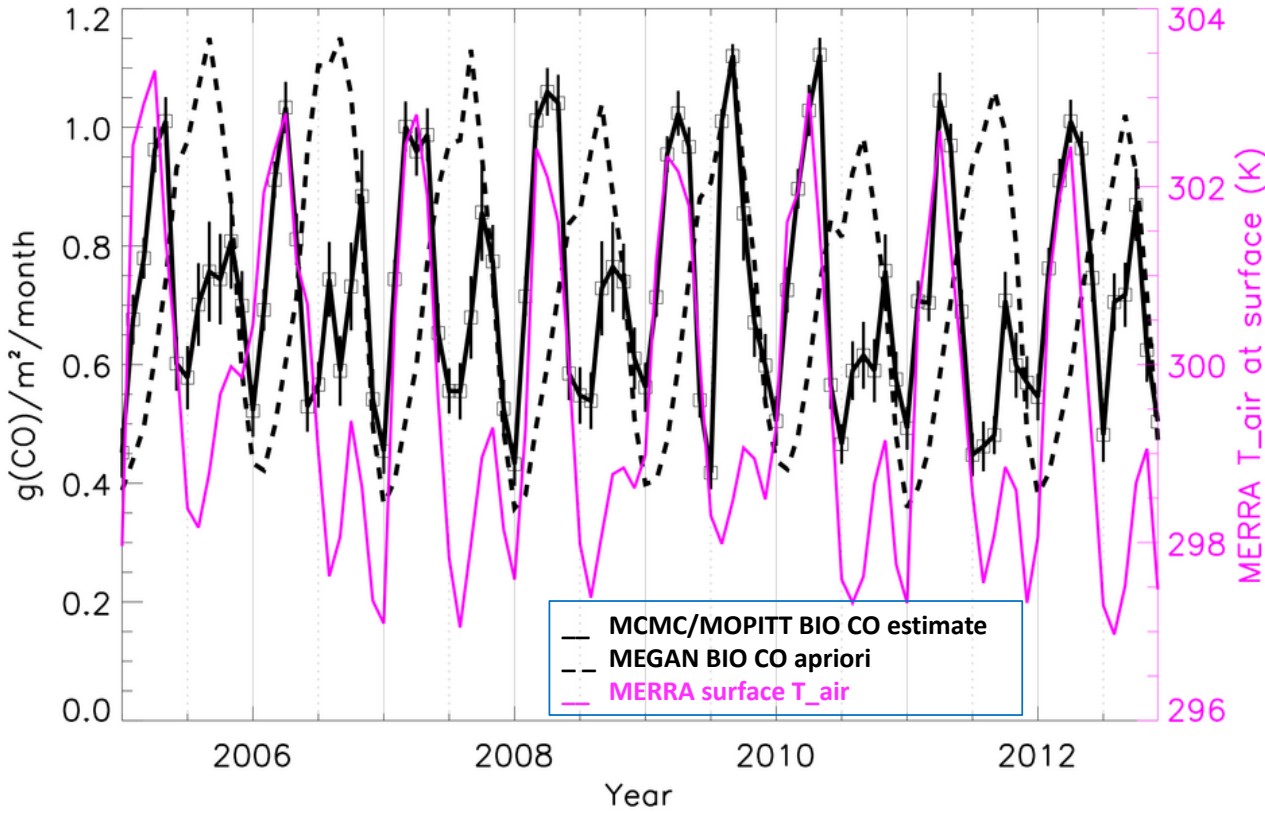

*Figure 3. Time series of a priori (dashed black) and posterior (solid black) CO fluxes with monthly mean 1-sigma errors and MERRA surface air temperature (magenta) for the N. Africa savannas region (see inset map in Fig. 2).*



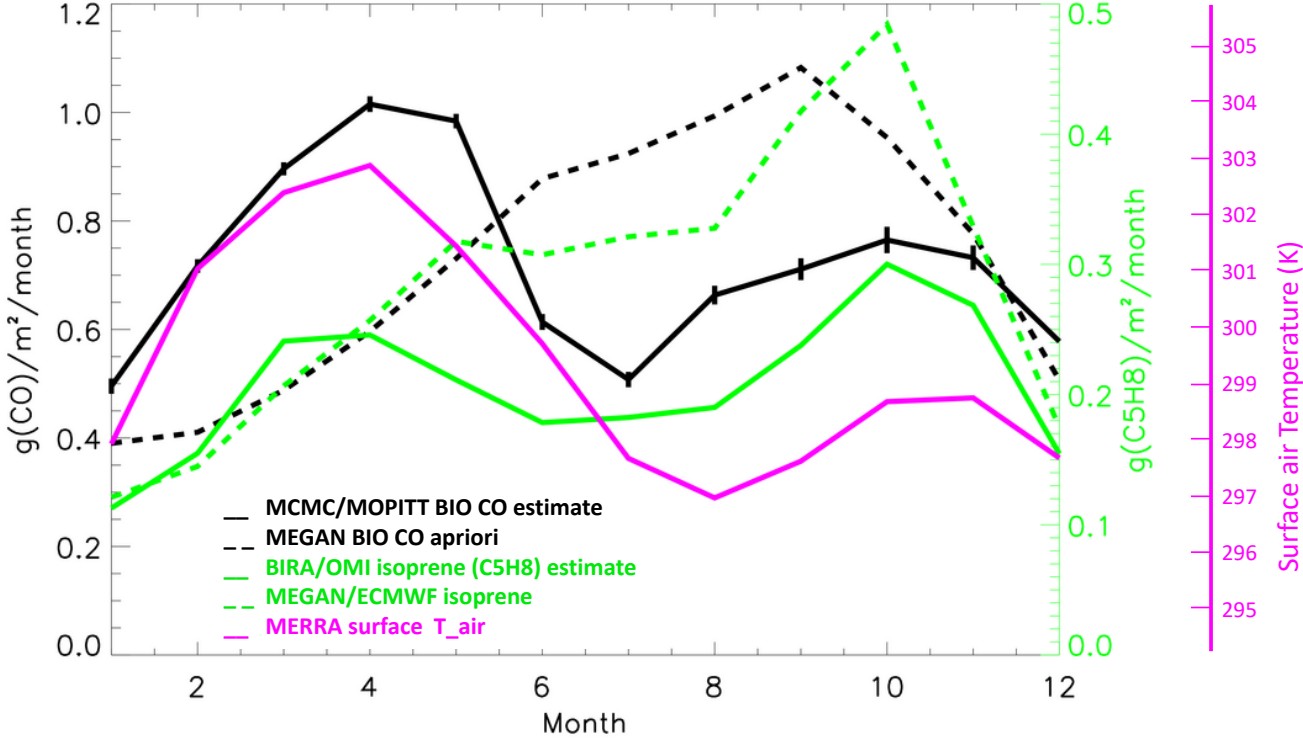

*Figure 4. Average monthly CO (black) and C5H8 (green) fluxes and surface air temperatures (magenta) for 2005-2012 for the N. Africa savannas region (see inset map in Fig. 2). Solid black and green lines show the posterior "top-down" fluxes while dashed black and green lines show the emissions predicted by MEGAN with associated meteorological fields.*

