# Peer review of "New Constraints on Biogenic Emissions using Satellite-Based Estimates of Carbon Monoxide Fluxes"

_Atmospheric Chemistry and Physics, 2019_

## Referee Comment (RC1) · Anonymous Referee #1 · 13 Jun 2019

Review of New Constraints on Biogenic Emissions using Satellite-Based Estimates of Carbon Monoxide Fluxes by Worden et al.

This paper deals with the top down estimate of biogenic CO emissions based on the GEOS-Chem model constrained with MOPITT observations. The paper brings interesting results about biogenic CO sources and their seasonal variability. The method provides improved estimation of these emissions. The paper is well structured, clear and well written. It should therefore be published in ACP. Nevertheless, the methodology and results that looks solid are often described too briefly. Some more detailed explanations should be given for some specific points that are detailed below.

[Figure]

P3: it is mentioned that 3 different MOPITT products are used (columns, full profiles and tropospheric profiles) to empirically evaluate errors due to transport. How is this error estimate integrated in the total error of the posterior fuxes? What are the error values?

P4: could you provide details about prior BB uncertainties? Some values?

P4: why 50% is assumed for BIO and FF prior flux estimates Is this value coming from sensitivity tests with varying uncertainties? Is this the value that provides the best fit between model and observations? This choice should be discussed as well as the metrics and methodology used to evaluate the improvement of the modeled CO distributions relative to the MOPITT observations. And the criteria used to decide that convergence is reached.

P5: the average posterior errors ar given. The different contributions to the error have been mentioned previously (such as the empirical transport error) but we do not have a clear idea about the complete budget. An equation indicating the different contributions to the posterior error and the contribution of each error source to the total error given here would be of interest.

P5: it is unclear to me why posterior error for FF is twice larger than for BIO and BB. I would have expected that this source is better constrained in the prior inventory. And why MOPITT constrain this source much less than the 2 others? Could the authors elaborate on this point?

P6: the present study finds BB emissions (290 Tg/yr) of about 1/3 of those from Folberth et al. 2006 (811 Tg/yr). It is a large difference that is briefly justified by the fact that tropical fires have declined during the 2005-2012 period relative to the one used in Folberth et al. 2006 according to Andela et al. (2017). Could you give more details to convince the reader ?

P7: how is the posterior estimate affected by the change in forcing fields (GEOS FP

versus GEOS-5? Is the top down method more robust to such changes than MEGAN?

P7: the results concerning the seasonality of the biogenic emissions are very interesting. The coincidence of isoprene and CO bimodal variability gives confidence in these results. Nevertheles, it is a bit desappointing not to have more explanations about the discrepancy between biogenic emissions and LAI variabilities! Are there some possible explanations? Why temperature plays a controling role in this N African Savannahs?

---

## Author Comment (AC1) · 13 Jul 2019

Review of New Constraints on Biogenic Emissions using Satellite-Based Estimates of Carbon Monoxide Fluxes by Worden et al.

This paper deals with the top down estimate of biogenic CO emissions based on the GEOS-Chem model constrained with MOPITT observations. The paper brings interesting results about biogenic CO sources and their seasonal variability. The method provides improved estimation of these emissions. The paper is well structured, clear and well written. It should therefore be published in ACP. Nevertheless, the methodology and results that looks solid are often described too briefly. Some more detailed explanations should be given for some specific points that are detailed below.

We thank the reviewer for their effort and useful comments and questions. We think that addressing these concerns will improve the manuscript. Our responses are embedded below in blue with modified or new text.

1) P3: it is mentioned that 3 different MOPITT products are used (columns, full profiles
and tropospheric profiles) to empirically evaluate errors due to transport. How is this error estimate integrated in the total error of the posterior fuxes? What are the error values?
Please see response to #4 below

2) P4: could you provide details about prior BB uncertainties? Some values?
Please see response to #4 below

3) P4: why 50% is assumed for BIO and FF prior flux estimates Is this value coming from sensitivity tests with varying uncertainties? Is this the value that provides the best fit between model and observations? This choice should be discussed as well as the metrics and methodology used to evaluate the improvement of the modeled CO distributions relative to the MOPITT observations. And the criteria used to decide that convergence is reached.

These uncertainties were chosen based on previous experience with error constraints and the objective of allowing the sector emissions to vary sufficiently to test new probability distributions within each grid cell. While a more complete sensitivity test would be desirable for future top-down inventory partitioning, our main goal for this manuscript was to demonstrate that this technique has skill in terms of reproducing the seasonal and spatial variability as found independently in top-down isoprene estimates using OMI HCHO observations. We will add the following to the text (section 3) where we state the use of ± 50%.

"This choice of uncertainty for the BIO and FF sectors is based on previous experience with error constraints and allows sufficient variability in the sector emissions for testing new probability distributions within each grid cell".

4) P5: the average posterior errors ar given. The different contributions to the error have been mentioned previously (such as the empirical transport error) but we do not have a clear idea about the complete budget. An equation indicating the different contributions to the posterior error and the contribution of each error source to the total error given here would be of interest.

P.4 Eq. 1 shows how the probability distribution is re-partitioned based on the errors assumed in each sector. This partitioning is unique for each grid cell and month so that a single equation showing error terms would not be very meaningful. We will include the following table and text in section 4 to help the reader understand the error sources and average outcomes for the tropical regions of interest in this study.

Table 1. Uncertainties applied in the Bayesian source attribution (Eq. 1). Values are monthly averages for single grid boxes (5° x 4° longitude x latitude) in the tropical study regions.

| CO sector distribution | A priori source | A priori uncertainty | Average Posterior Uncertainty (tropics grid boxes) |
|---|---|---|---|
| Total flux (top-down estimate) | Jiang et al., (2017) Inversion based on MOPITT CO data | ± 50% (assumed) | ± 12% average constraint[a], with 11% 1-sigma standard deviation for tropical grid cells[b] |
| BB (biomass burning) | GFED4s (van der Werf et al., 2017) | ± 24% (Akagi et al., 2011) | ± 22% |
| BIO (biogenic NMVOCs) | MEGAN v2.0 (Guenther et al., 2006) | ± 50% (assumed) | ± 24% |
| FF (fossil fuels) | EDGAR 3.2 Olivier and Berdowski, 2001 | ± 50% (assumed) | ± 45% |

[a] The total flux posterior error is estimated from 3 flux inversion types (see text for description) to approximately account for model transport errors.
[b] Average and standard deviation are computed for tropics (20°S to 20°N) using grid boxes with with emissions > 0.1 gCO/m2/month.

"Uncertainties are available by 5° x 4° grid cell, month and source sector (BB, FF or BIO) and represent the 1-sigma width of the posterior distributions; these distributions are critically dependent on the a priori uncertainties and therefore subject to change when different a priori distributions and covariances are assumed in the Bayesian attribution approach. Table 1 lists the sources of a priori data and uncertainties and gives average monthly values representative of the

individual grid cells used in this study. For the remote tropical regions considered here, FF contributions to total CO fluxes are small and we find the most improvement over prior errors in BIO CO posterior flux uncertainties, especially in months with little or no BB emissions. This can be seen in Fig. 2, where monthly grid box posterior errors were averaged spatially for the region of interest and over years 2005-2012. One of the assumptions in this study is the prior uncertainty in BB, which only considers emission factor uncertainties (Akagi et al., 2011) and does not explicitly account for other factors in BB CO fluxes such as combustion completeness and biomass (fuel) amount (e.g. Bloom et al. 2015). Future work will examine the effects of using a wider range of prior uncertainties that reflect multiple inventories."

5) P5: it is unclear to me why posterior error for FF is twice larger than for BIO and BB. I would have expected that this source is better constrained in the prior inventory. And why MOPITT constrain this source much less than the 2 others? Could the authors elaborate on this point?

The FF component is very small in the tropical regions we consider so there is little information to improve on the FF error compared to the prior. This was already stated in the text, but revisions to address the comment above (e.g., error table) make this more explicit.

6) P6: the present study finds BB emissions (290 Tg/yr) of about 1/3 of those from Folberth et al. 2006 (811 Tg/yr). It is a large difference that is briefly justified by the fact that tropical fires have declined during the 2005-2012 period relative to the one used in Folberth et al. 2006 according to Andela et al. (2017). Could you give more details to convince the reader ?

This could also be an overestimation in the BB CO emissions considered by Folberth et al (2006). Recent estimates using GFED4 (van der Werf et al., 2017) report annual mean emissions for the 1997-2016 period for CO as 357 Tg/yr, while Granier et al., 2011 reported a range of 414 to 509 Tg/yr for 6 inventories in the 1997-2000 period, a period with significant interannual variability due to the strong 1997-1998 ENSO episode.

We will modify the text to state:

This contribution from BIO CO represents a larger percentage (~41%) of the sum of BB, FF and BIO CO sources than expected (~27%) based on Folberth et al. (2006) which has 811 Tg(CO)/yr for BB and 672 Tg (CO)/yr for FF). However, there is a wide range in reported biomass burning emission estimates, with large interannual variability. van der Werf et al., (2017) report 357 Tg/yr mean emissions for BB CO over 1997-2016 while Granier et al., (2011) reported a range of 414 to 509 Tg/yr for 6 emission inventories in the 1997-2000 period. Because our 2005-2012 study period did not include the significant ENSO episodes in 1997 and 2015, we would expect lower average values for BB CO emissions. Furthermore, in recent decades, there is a decreasing contribution of BB CO associated with a decline in tropical fires (e.g., Andela et al., 2017), as

well as declining FF CO emissions (Yin et al., 2015; Strode et al., 2016; Jiang et al., 2017; Zheng et al., 2018).

7) P7: how is the posterior estimate affected by the change in forcing fields (GEOS FP versus GEOS-5? Is the top down method more robust to such changes than MEGAN?

Although there does appear to be less dependence on the version of the meteorological fields in the posterior results compared to the MEGAN apriori (see Response Figure 1 below for the N. African savannas region), we did not want to draw conclusions, especially about trends in biogenic fluxes, without more consistent meteorological fields. Also, the dependence on the prior can still vary from region to region depending on the errors in the other emission terms (BB and FF). This is more obvious in Response Figure 2 for the Equatorial Africa region where there is more interference from BB emissions, and the time dependence of the prior is more clearly affecting the posterior result. Therefore, we chose the 2005-2012 period for the analysis in this paper.

8) P7: the results concerning the seasonality of the biogenic emissions are very interesting. The coincidence of isoprene and CO bimodal variability gives confidence in these results. Nevertheles, it is a bit desapointing not to have more explanations about the discrepancy between biogenic emissions and LAI variabilities! Are there some possible explanations? Why temperature plays a controling role in this N African Savannahs?

Marais et al., (2014) originally found that surface layer temperature dominates over LAI for controlling isoprene emissions in the N. African savannas. They hypothesized that since LAI is less than 2.5 m2 m-2 year round, MEGAN dependence is not saturated. At the same time, MODIS observations could underestimate LAI during the rainy season due to cloud contamination. This reference is already cited. Furthermore, the study presented here is meant to demonstrate the methods that we will build on in future work to test the processes and potential changes needed in MEGAN to reproduce top-down estimates of biogenic emissions.

[Figure]

*Response figure 1. Timeseries of apriori (dashed blue) and estimated CO flux (solid black, with error bars) for the N. African savannas region. Green arrows indicate the different time periods for the GEOS-4, GEOS-5 and GEOS-FP meteorological fields used to calculate the apriori with the MEGAN model and for the inverse analysis for total CO flux.*

[Figure]

*Response figure 2. Timeseries of apriori (dashed blue) and estimated CO flux (solid black, with error bars) for the Equatorial Africa region. Green arrows indicate the different time periods for the GEOS-4, GEOS-5 and GEOS-FP meteorological fields used to calculate the apriori with the MEGAN model and for the inverse analysis for total CO flux.*

References to be added:

van der Werf, G. R., Randerson, J. T., Giglio, L., van Leeuwen, T. T., Chen, Y., Rogers, B. M., Mu, M., van Marle, M. J. E., Morton, D. C., Collatz, G. J., Yokelson, R. J., and Kasibhatla, P. S.: Global fire emissions estimates during 1997–2016, Earth Syst. Sci. Data, 9, 697-720, https://doi.org/10.5194/essd-9-697-2017, 2017.

Granier, C., Bessagnet, B., Bond, T. et al.: Evolution of anthropogenic and biomass burning emissions of air pollutants at global and regional scales during the 1980–2010 period, Climatic Change (2011) 109: 163. https://doi.org/10.1007/s10584-011-0154-1

---

## Referee Comment (RC2) · Anonymous Referee #2 · 9 Aug 2019

Comment on : "New Constraints on Biogenic Emissions using Satellite-Based Estimates of Carbon Monoxide Fluxes " By Helen Worden et al.

The paper "New Constraints on Biogenic Emissions using Satellite-Based Estimates of Carbon Monoxide Fluxes" provides an improved estimation of the biogenic emission, comparing model simulation based on Bottom up inventories with a satellite based "Top Down" emission estimation for CO. The CO production from biogenic emissions (BIO), together with Biomas Burning (BB) and Fossil Fuel (FF) consumption is one of the three most important parts of the CO budget and Flux (F). The "Top down" estimate provides an estimate of the Total CO2 Flux (F) without the ability to distinguish

the individual sources and sectors, but in this work the information of the total flux is used to improve the estimate of the biogenic emissions, just using the Bayes probabilities approach. The new approach is realized individually for each grid cells of $4°x5°$ and month. A systematic pattern and spatial distribution is obtained and compared to other measurements. 1) Biogenic emissions of the isoprene retrieved from the OMI instrument shows a very similar distribution. 2) The temporal pattern which shows a significant difference between apriori and posteriori biogenic CO flux for the north African Savanna is studied and compared to the surface temperature.

General comment:

The Work is well written, interesting and matches the scope of ACP, it should be published after minor correction and after including a bit more information about the methodology. At the moment the paper is quite compact with just one example (region), but the supplement provides more examples, which is adequate and a good idea.

The new of the paper is that it somehow combines a model study and therefore a detailed "Bottom up" estimation, which contain a detailed distribution of different sectors (BIO,BB,FF) together with a satellite based "Top Down" approach which, just report the total flux "F", latter is somehow a measurement, while the prior is the apriori information.

Unfortunately the description is very short and the approach cannot be easily be reproduced.

I imagine that the implementation of the Baysian approach ends up in a least square fitting equation and looking finally for the minimum of something like the following cost function will help to find the posterior solution: $1/\sigma2$ (F(BIO,BB,FF)-A)ˆ2 + (([BIO,BB,FF]-xapr)ˆT (S_(BIO,BB,FF))ˆ(-1)([BIO,BB,FF]-xapr)

with $\sigma$ the uncertainty in the "Top down" approach A , F= the total Flux or Forward

Modell F= CH4 related part + BIO+BB+FF. S_(BIO,BB,FF) might be the more or less diagonal covariance matrix which describe the uncertainty.

If it is some how different, it would be nice to get an more easily insight in the criteria which equation is used to determine the vector BIO,BB,FF. Specific comments: 3 Bayesian CO flux attribution approach I think, this a very crucial section for the work and unfortunately not very easy to understand.

ðİŚİ(BB,BIO,FF|A) âĹİ ðİŚİ(BB,BIO,FF) p(F|A)/p(F) Eq. 1.

I understand that :

ðİŚİ(BB,BIO,FF|A)   âĹİ   ðİŚİ(BB,BIO,FF)   p(A|   BB,BIO,FF)/P(A)   and p(F,A)=p(F|A)p(A)=p(A|F) p(F) and probable it is valid that P(A|F)=P(A| BB,BIO,FF) as P(F| BB,BIO,FF) =1.0. But here it would be helpful to get a bit more info, and define the relation between F and (BIO,BB,FF).

Where I get a bit problems is with the statement p(F) = 50%, does this mean p(F)=0.5 As F is a continuous quantity p(F) might be a probability density function pdf and it should be something like p(F) dF = 0.5. Or more likely it should say p(F) is a Gaussian distribution with a priori Fapriori as most probable, mean value and sigma as stdv .

p(F)= 1/sqrt(2 Pi sigma**2) exp(- ((F-Fapriori)/sigma )**2) and sigma=0.5*Fapriori

Or is the pdf a more general pdf, which is produced by the (MCMC) algorithm. If latter is the case, it would be nice to get somehow the formula of the a posteriori estimation, finally it should just be an weighted mean between the three a priori informations BIO,BB,FF aprioris and their a priori Stdev and the Top down estimation of their sum.

Similar might apply for other uncertainties and pdf as p(F|A). I would assume that it is assumed to be Gaussian and the standard deviation is calculated from the ensemble of three "top down" inversion estimates, but up to now this is not described clearly.

Same the different between F and A, is not be explained. Please include the equations

how F is calculated as function of BIO,BB,FF and 877 Tg/yr, at least in the supplement.

So far I understood the methodology, the estimated BIO, BB and FF (the solution vector x) is an optimal estimation. The finally reconstructed BIO, BB and FF emission in each grid cell, matches more or less their a priories and explain more or less the "Total flux of CO" (which is some how the measurement y), which is their minus some fix parts as CO from CH4 (F-rest ) matches the "Top down" estimation of (F|A). The authors recommend to read the description of another earlier publication, but I would recommend amplify the description at least a little bit, as the method is very crucial for the work. And maybe why not use the supplement document to write down the complete mathematical expression, which would allow to reproduce the approach.

4 Uncertainty prediction and limitations

The use of a measured total flux and redistribute the fluxes of the different sectors, might produce a very strong dependence between the errors in BIO,BB,FF. Is there a way to characterize this ? How could the estimate improve, if you could reduce the uncertainty in FF to 0.0 .

One of the main results is the very nice correlation between Surface Temperature and BIO-Emission: The CO flux "Top Down" estimation is based on the joint near and also mid infrared MOPIIT retrieval product. The result and sensitivity of mid infrared nadir sensors might depend on the surface temperature. Therefore it would be nice to discuss shortly if such errors could be relevant.

6 Global budgets of CO and C5H8 from biogenic emissions

Maybe it would be nice to see an correlation plot between OMI based C5H8 and a) the apriori and b) a posteriori estimated biogenic CO flux.

7 Seasonality of biogenic emissions – case study for the North African Savannas

As mentioned earlier, just for the completeness it would be nice just to discuss if the Surface Temperature or other surface properties which might have an impact on the

CO MOPIIT retrieval.

Table1: Maybe could you include "F" or "A" in this table. Suggestion: the "MEGAN" emission estimate is the apriori and might be included in the same box just in brackets together with the apriori uncertainty .

---

## Author Comment (AC2) · 6 Sep 2019

We have updated our replies to include some of the changes and other author comments received in addressing referee #2.

Please also note the supplement to this comment: https://www.atmos-chem-phys-discuss.net/acp-2019-377/acp-2019-377-AC2-supplement.pdf

---

## Author Comment (AC3) · 6 Sep 2019

Review of New Constraints on Biogenic Emissions using Satellite-Based Estimates of Carbon Monoxide Fluxes by Worden et al.

Anonymous Referee #2

The paper "New Constraints on Biogenic Emissions using Satellite-Based Estimates of Carbon Monoxide Fluxes" provides an improved estimation of the biogenic emission, comparing model simulation based on Bottom up inventories with a satellite based "Top Down" emission estimation for CO. The CO production from biogenic emissions (BIO), together with Biomas Burning (BB) and Fossil Fuel (FF) consumption is one of the three most important parts of the CO budget and Flux (F). The "Top down" estimate provides an estimate of the Total CO2 Flux (F) without the ability to distinguish the individual sources and sectors, but in this work the information of the total flux is used to improve the estimate of the biogenic emissions, just using the Bayes probabilities approach. The new approach is realized individually for each grid cells of 4°x5° and month. A systematic pattern and spatial distribution is obtained and compared to other measurements. 1) Biogenic emissions of the isoprene retrieved from the OMI instrument shows a very similar distribution. 2) The temporal pattern which shows a significant difference between apriori and posteriori biogenic CO flux for the north African Savanna is studied and compared to the surface temperature.

General comment:
The Work is well written, interesting and matches the scope of ACP, it should be published after minor correction and after including a bit more information about the methodology. At the moment the paper is quite compact with just one example (region), but the supplement provides more examples, which is adequate and a good idea.

*We thank the reviewer for their effort and useful comments and questions. We think that addressing these concerns will improve the manuscript. Our responses are embedded below in blue italics with modified or new text.*

1) The new of the paper is that it somehow combines a model study and therefore a detailed "Bottom up" estimation, which contain a detailed distribution of different sectors (BIO,BB,FF) together with a satellite based "Top Down" approach which, just report the total flux "F", latter is somehow a measurement, while the prior is the apriori information. Unfortunately the description is very short and the approach cannot be easily be reproduced. I imagine that the implementation of the Baysian approach ends up in a least square fitting equation and looking finally for the minimum of something like the following cost function will help to find the posterior solution: $1/\sigma^2$ (F(BIO,BB,FF)-A)^2 + (([BIO,BB,FF]-xapr)^T (S_(BIO,BB,FF))^(-1)([BIO,BB,FF]-xapr) with $\sigma$ the uncertainty in the "Top down" approach A, F= the total Flux or Forward Modell F= CH4 related part + BIO+BB+FF. S_(BIO,BB,FF) might be the more or less diagonal covariance matrix which describe the uncertainty. If it is some how different, it would be nice

to get an more easily insight in the criteria which equation is used to determine the vector BIO,BB,FF.

*We have expanded the discussion in section 3 to include explicit details of the probability distributions and cost function with added equations 3-5, and a priori uncertainties given in the new Table 1. We agree with the reviewer that these additions improve the reproducibility of this study.*

Specific comments:

2) 3 Bayesian CO flux attribution approach I think, this a very crucial section for the work and unfortunately not very easy to understand. ðISI(BB,BIO,FF|A) âĹİ ðIṠI(BB,BIO,FF) p(F|A)/p(F) Eq. 1.

   I understand that :
   ðISI(BB,BIO,FF|A)âĹİðIṠI(BB,BIO,FF) p(A|BB,BIO,FF)/P(A) and
   p(F,A)=p(F|A)p(A)=p(A|F) p(F) and probable it is valid that P(A|F)=P(A|BB,BIO,FF) as P(F| BB,BIO,FF) =1.0. But here it would be helpful to get a bit more info, and define the relation between F and (BIO,BB,FF).

   Where I get a bit problems is with the statement p(F) = 50%, does this mean p(F)=0.5 As F is a continuous quantity p(F) might be a probability density function pdf and it should be something like p(F) dF = 0.5. Or more likely it should say p(F) is a Gaussian distribution with a priori Fapriori as most probable, mean value and sigma as stdv. p(F)= 1/sqrt(2 Pi sigma**2) exp(- ((F-Fapriori)/sigma )**2) and sigma=0.5*Fapriori
   Or is the pdf a more general pdf, which is produced by the (MCMC) algorithm. If latter is the case, it would be nice to get somehow the formula of the a posteriori estimation, finally it should just be an weighted mean between the three a priori informations BIO,BB,FF aprioris and their a priori Stdev and the Top down estimation of their sum. Similar might apply for other uncertainties and pdf as p(F|A). I would assume that it is assumed to be Gaussian and the standard deviation is calculated from the ensemble of three "top down" inversion estimates, but up to now this is not described clearly. Same the different between F and A, is not be explained. Please include the equations how F is calculated as function of BIO,BB,FF and 877 Tg/yr, at least in the supplement.

*The explicit probability distributions and cost function are now added as equations 3-5. Since these apply to each model grid cell, the global 877 Tg/yr of CO from CH4 destruction is not estimated in the MCMC approach. This is a global constraint determined by Jiang et al., (2017).*

3) 4 Uncertainty prediction and limitations
   The use of a measured total flux and redistribute the fluxes of the different sectors, might produce a very strong dependence between the errors in BIO,BB,FF. Is there a way to characterize this ? How could the estimate improve, if you could reduce the uncertainty in FF to 0.0.

*We have added a new Table 1 in response to Ref. #1 that shows how aposteriori uncertainties are reduced compared to the prior assumptions. While there will be a dependence on the relative errors in BIO, BB, FF in general, for the tropical regions studied here, the FF emissions are small enough (Figure 2) that even reducing the prior uncertainty to 0.0 would not make a significant difference in the BIO term. Sensitivity to the choice of prior uncertainty will be the subject of future work.*

4) One of the main results is the very nice correlation between Surface Temperature and BIO-Emission: The CO flux "Top Down" estimation is based on the joint near and also mid infrared MOPIIT retrieval product. The result and sensitivity of mid infrared nadir sensors might depend on the surface temperature. Therefore it would be nice to discuss shortly if such errors could be relevant.

*Sensitivity in mid-infrared nadir sensor retrievals depends on thermal contrast between the surface and atmosphere. In the case of vegetated tropical biomes, thermal contrast near the surface is usually close to zero, regardless of surface temperature, due to humidity. For the MOPITT joint thermal and near-IR product, sensitivity to near surface CO in these regions is mostly driven by NIR surface albedo (Worden et al., 2010). Furthermore, variations in the sensitivity of the MOPITT retrievals are characterized by the averaging kernels which were included when the data were assimilated for the original flux estimates in Jiang et al.(2017). Therefore, we don't expect the dependence of MOPITT vertical sensitivity on surface temperature to be relevant to the re-partitioned flux estimates. However, since MOPITT retrievals require cloud free observations, there could be inflated errors in the flux estimates due to fewer MOPITT samples during the rainy season, which corresponds to lower surface temperatures. This effect would need to be included in a future study with more comprehensive flux errors. The new Table 1 in section 4 (in response to referee #1) now includes the variance of tropical grid cell posterior flux errors, and we will add a footnote into the new Table 1:*

"The variance in tropical grid cell flux errors includes both spatial and temporal variability, however, these errors have not been weighted to account for sampling effects, such as inflated errors due to fewer MOPITT observations during rainy seasons".

5) 6 Global budgets of CO and C5H8 from biogenic emissions
Maybe it would be nice to see an correlation plot between OMI based C5H8 and a) the apriori and b) a posteriori estimated biogenic CO flux.

*We looked at spatial and temporal correlations of isoprene with apriori (isoprene and CO) and aposteriori CO and could not find conclusive results. This is likely due to the high correlation for low-emission grid cells. Overall, MEGAN is usually too high for both C5H8 and CO, which is what we are trying to convey in the table.*

6) 7 Seasonality of biogenic emissions – case study for the North African Savannas

As mentioned earlier, just for the completeness it would be nice just to discuss if the Surface Temperature or other surface properties which might have an impact on the CO MOPIIT retrieval.

*Please see above response to comment #4. We think the revisions to section 4 now give a more comprehensive description of uncertainties that would not benefit from further discussion in section 7.*

7) Table1: Maybe could you include "F" or "A" in this table. Suggestion: the "MEGAN" emission estimate is the apriori and might be included in the same box just in brackets together with the apriori uncertainty.

*Thank you for this suggestion. This simplified the table and also allowed us to include more information, i.e., the apriori values for the BB and FF sectors. We also found that the posterior uncertainties in the tropical sub-regions needed to be corrected by sqrt(8) to account for the 8-year average. This was already reported correctly in the other regions.*

REFERENCES:

Worden, H. M., M. N. Deeter, D. P. Edwards, J. C. Gille, J. R. Drummond, and P. Nédélec (2010), Observations of near-surface carbon monoxide from space using MOPITT multispectral retrievals, *Journal of Geophysical Research (Atmospheres)*, *115*(d14), 18314, doi:10.1029/2010JD014242.